# Weight-Adjusted Waist Index (WWI)—A Promising Anthropometric Indicator of Depressive Symptoms in Hospitalized Older Patients

**DOI:** 10.3390/nu17010068

**Published:** 2024-12-28

**Authors:** Renata Korycka-Bloch, Pawel Balicki, Agnieszka Guligowska, Bartlomiej K. Soltysik, Tomasz Kostka, Zuzanna Chrzastek

**Affiliations:** Department of Geriatrics, Healthy Ageing Research Centre, Medical University of Lodz, 92-209 Lodz, Poland; renata.korycka-bloch@umed.lodz.pl (R.K.-B.); pawel.balicki@umed.lodz.pl (P.B.); agnieszka.guligowska@umed.lodz.pl (A.G.); bartlomiej.soltysik@umed.lodz.pl (B.K.S.); tomasz.kostka@umed.lodz.pl (T.K.)

**Keywords:** WWI, BMI, depressive symptoms, older adults

## Abstract

Objectives: The aim of this study was to evaluate which anthropometric index, either body mass index (BMI) or weight-adjusted waist index (WWI), is more accurately associated with the prevalence of the most common chronic diseases and components of geriatric assessment in hospitalized older adults. Methods: The study included a total of 2945 hospitalized older adults (median age 82 years). The associations between the presence of chronic diseases and Comprehensive Geriatric Assessment (CGA) results were compared with WWI and BMI values. Results: The WWI was significantly higher in both sex groups suffering from hypertension, diabetes, osteoarthritis, and depression. In women, the parameter was increased among individuals with previous myocardial infarction, who presented heart failure symptoms or had chronic kidney disease diagnosed, whereas in men, among those with pulmonary diseases and osteoporosis, WWI was related to many CGA parameters oftentimes where BMI proved to fail. There was a positive correlation of WWI with the presence of depressive symptoms assessed with the geriatric depression scale (GDS) but no significant correlation with BMI. In multiple logistic regression models, WWI was a stronger predictor of depression as compared to waist circumference or the waist-to-height ratio. Conclusions: There is an association between a higher WWI and depression diagnosis as well as the presence of depressive symptoms according to the GDS in hospitalized older adults, both women and men. There is no such correlation between depression and BMI. Both high BMI and high WWI values seem to identify older patients with cardiometabolic diseases such as hypertension and diabetes. According to this study, WWI seems to be a promising indicator of depression risk and, similarly to BMI, a useful parameter for the assessment of cardiometabolic risk in older hospitalized adults.

## 1. Introduction

In recent decades, there has been a significant worldwide increase in the number of individuals aged 60 years and over, particularly in developed countries. The World Health Organization (WHO) projects that by 2030, one in six people globally will be aged 60 years or over. This trend is generally positive as long as it aligns with the concept of ‘successful ageing’, which can be achieved through the maintenance of optimal physical and mental health, as well as appropriate prevention and treatment of diseases commonly associated with older adults [1]. Unfortunately, a substantial part of the older population suffers from multimorbidity (MM), which severely impacts their quality of life [2]. Obesity is a major risk factor for the development of MM, and the prevalence of obesity continues to rise worldwide [3,4]. Obesity is typically diagnosed based on the Body Mass Index (BMI), and the incidence of numerous chronic diseases, such as hypertension, diabetes, dyslipidemia, and metabolic syndrome, increases substantially with a higher BMI [5,6]. However, BMI has several limitations, particularly in older populations. Older adults are often characterized by reduced height, decreased muscle mass, and increased fat mass percentage, which suggests that other anthropometric measurements should also be considered [7]. Normal weight obesity (NWO), defined as a normal BMI (<25 kg/m^2^) but with a high percentage of body fat, is an emerging concern in older adults as it leads to physical impairments [8]. The literature findings suggest that older adults with NWO are at an increased risk of sarcopenia, emphasizing the need for more detailed anthropometric assessment as part of screening in this population [9]. A large study involving 177,792 participants suggests that, though remaining undetected in BMI-based screening, normal weight obesity increases the prevalence of cardiometabolic risk factors [10]. Similarly, sarcopenic obesity, defined as the loss of muscle tissue and muscle strength coupled with an accumulation of body fat, is increasingly prevalent in older populations and exacerbates physical functional impairments and disabilities [11]. To assess sarcopenia or sarcopenic obesity, muscle mass evaluation is necessary; however, comprehensive body composition assessments are often impossible due to a lack of appropriate equipment [12]. Therefore, there is an urgent need in geriatrics for alternative anthropometric indices, other than BMI, that could be widely used to describe the older population.

The weight-adjusted waist index (WWI), defined as waist circumference (WC) divided by the weight in kilograms squared, has been shown to be a strong predictor of several chronic diseases, including hypertension, diabetes, cardiovascular diseases, chronic kidney disease, and albuminuria, as well as a predictor of mortality [13,14,15]. WWI, as a promising new indicator, utilizes WC measurement and proves to be a more reliable predictive factor than BMI, especially in the case of central obesity [16,17]. Furthermore, WWI is positively associated with fat mass and negatively associated with muscle mass in older adults, reflecting the morphology of sarcopenic obesity [17,18]. Recent studies have suggested that among anthropometric indices, WWI has the strongest association with sarcopenic obesity in older populations [19,20]. Sarcopenic obesity is closely connected with frailty syndrome as both conditions share overlapping features and mechanisms that contribute to their development and progression in older adults [21]. Emerging research also suggests that WWI may serve as a potential predictive marker for frailty syndrome [22,23]. The latest study conducted by Jia et al. highlights the importance of WWI as an anthropometric parameter independent of BMI, associated with accelerated aging [24]. Nonetheless, there remains a notable lack of studies evaluating WWI in large cohorts of vulnerable older adults.

It remains unclear whether the relationship between anthropometric indices and cardiometabolic differs in advanced-age individuals and what their association with typical geriatric conditions such as functional and nutritional status, dementia, or depression is. Therefore, the aim of this study was to establish which anthropometric index, BMI or WWI, is more accurately associated with the most common chronic diseases and components of geriatric assessment in older hospitalized populations.

## 2. Materials and Methods

### 2.1. Participants

The study included a total of 2945 older adults hospitalized in the Department of Geriatrics of the Medical University of Lodz in the years 2012–2023. The dominance of women (*n* = 2095) over men (*n* = 850) in the studied group results from the profile of Polish demography. The inclusion criteria were the following: ≥60 years of age and the ability to undergo the Comprehensive Geriatric Assessment (CGA). The participants who could give informed consent themselves (i.e., those with the cognitive ability to understand the study purpose, procedures, and potential risks) were asked to submit the informed consent form. Those who were unable to provide informed consent due to cognitive impairment (such as severe dementia or other diseases that hinder verbal and logical contact) were excluded from the study.

### 2.2. Anthropometric Measurements

The study participants were weighed and measured on a RADWAG personal weight scale (WPT60 150OW) (RADWAG Balances and Scales, Radom, Poland). Waist (WC), calf (CC), and arm (AC) circumferences were measured using a SECA measuring tape (SECA Deutschland, Hamburg, Germany). BMI (body weight [kg]/height [m]^2^) and the waist-to-height ratio (WHtR) (WC [cm]/height [cm]) were calculated. The weight-adjusted waist index (WWI) was calculated by dividing WC by the square root of the body weight [cm/√kg].

### 2.3. Comprehensive Geriatric Assessment

A comprehensive geriatric assessment (CGA) was conducted by medical doctors on admission to the hospital. The procedure is defined as a multidisciplinary diagnostic process that identifies the functional, psychological, social, and medical capabilities of an older adult. This examination is vital for the planning of care, rehabilitation, and treatment of older patients to optimize further management. CGA includes tests evaluating functional efficiency, mental health, physical status, and social issues. The Vulnerable Elders Survey (VES-13) was conducted to assess the risk of health deterioration in the studied population [25].

The Activity of Daily Living (ADL) and Instrumental Activities of Daily Living (IADL) scales were used to assess the functional status of the participants. ADL, a scale proposed by Sidney Katz, includes common activities of daily living such as feeding, bathing, clothing, using the toilet, moving, and controlling sphincter function. The patient receives one point for each affirmative answer. The maximum number of points is six. The higher the score obtained, the better the patient’s functional status [26]. Another scale, IADL, developed by Lawton and Brody, is used to assess the range of activities that support independent living such as the ability to use the telephone, shopping, food preparation, housekeeping, laundry, mode of transportation, responsibility for own medication, and ability to handle finances. A summary score ranges from zero to eight points (from dependency to high functional status, respectively) [27].

Each patient’s mental status was assessed with the mini mental state examination (MMSE), the clock-drawing test (CDT), and the geriatric depression scale (GDS). MMSE is a 30-point questionnaire used to measure the patient’s cognitive impairment by examining functions such as orientation to time and place, attention and calculation, short-time memory, language skills, and visuospatial abilities. The maximum score is 30 points, and ≥25 points correspond to normal cognition. A score below 24 indicates possible cognitive impairment [28,29]. CDT is a screening tool for dementia used to assess executive function, visual–spatial ability, motor programming, attention, and concentration. A score of ten points suggests proper cognitive functioning, whereas a score below five points indicates prominent impairment [30]. The presence of depression symptoms and their severity were assessed with the GDS scale, which includes 15 simple yes/no alternative questions. The highest possible score is 15, and a result >5 indicates the presence of depression symptoms [31].

Nutritional status was assessed with the Nutritional Risk Screening (NRS). NRS is a tool used to detect any existing malnutrition or risk of malnutrition in hospitalized patients, taking into account their BMI, weight loss within three months, reduced dietary intake in the last week, and severe illness. The total score ranges from zero to seven. A score > 3 indicates the patient’s improper nutritional status [32].

Age and chronic diseases of the study participants were also recorded.

### 2.4. Statistical Analysis

The normality of data distribution was verified by the Shapiro–Wilk test. Since the data were non-normally distributed, the variables were presented as median values and interquartile ranges. Quantitative variables were compared between the sex groups using the Mann–Whitney U test. The Chi^2^ test was used to compare qualitative values, i.e., the presence or absence of chronic diseases. Spearman’s rank correlation coefficients were calculated to measure the association between quantitative values and WWI and BMI. The Mann–Whitney U test was applied to compare qualitative data (absence or presence of chronic diseases) according to median values and calculate interquartile ranges of WWI and BMI (separately for men and women). Multiple logistic regression models were performed with WWI, WC, WHtR, age, and sex as independent variables and depression as a dependent variable. Odds ratios (OR) and 95.0% confidence intervals were calculated. The limit of statistical significance was set at a *p*-value of less than 0.05. The analyses were carried out using the Statistica 13.3 software (StatSoft Polska, Cracow, Poland).

## 3. Results

Table 1 presents a summary of the general characteristics of the study population categorized by sex. The men exhibited a higher body weight and had larger WC, CC, and AC compared to the women. Conversely, the women demonstrated a higher WHtR and WWI than the men. The GDS and VES-13 scores were significantly higher in the female than in the male group. Performance on the CDT was superior in the latter. The prevalence of dyslipidemia, osteoarthritis, osteoporosis, and depression was higher in the female patients, whereas diabetes, myocardial infarction, atrial fibrillation, and Parkinson’s disease were more prevalent in the male subjects (Table 1).

Table 2 presents Spearman’s rank correlation coefficient between quantitative variables and the WWI index. The results indicate a positive correlation between BMI, WC, CC, AC, WHtR, and GDS with the WWI index in both women and men. In the female group, age and VES-13 showed a positive correlation, while ADL, IADL, and MMSE demonstrated negative correlations with WWI. In the male group, CDT results were positively correlated with WWI (Table 2).

Table 3 presents the Spearman’s rank correlation coefficient between quantitative variables and BMI. The results indicate a positive correlation between weight, WWI, WC, CC, AC, WHtR, MMSE, and CDT with BMI in both women and men. Age, VES-13, and NRS were negatively correlated with BMI in both sexes. In the female group, ADL and IADL showed a positive correlation with BMI. In the male group, GDS was negatively correlated with BMI (Table 3).

Table 4 presents a comparison between the qualitative data (chronic diseases) and WWI. The median and interquartile range were calculated separately for the women and men, both for the groups with and without disease. WWI was significantly higher in both sexes for individuals suffering from hypertension, diabetes, osteoarthritis, and depression. Additionally, in the female group, WWI was higher in those with previous myocardial infarction and who had heart failure or chronic kidney disease. While in the male group, those with pulmonary diseases and osteoporosis had a higher WWI compared to individuals without these diseases (Table 4).

Table 5 presents a comparison between qualitative data (chronic diseases) and BMI. The median and interquartile range were calculated separately for the women and men, for both groups with and without disease. The BMI was significantly higher in both sexes among individuals suffering from hypertension, dyslipidemia, diabetes, and osteoarthritis. Additionally, in the female group, BMI was higher in patients with atrial fibrillation and heart failure. Both women and men with a history of cancer, osteoporosis, and dementia had a lower BMI compared to those without these conditions. In the male group, those with pulmonary diseases had a lower BMI than individuals not affected by these conditions (Table 5).

Finally, as WWI is highly correlated with WC and WHtR, the result was compared with WC and WHtR to verify their predictive value for the presence of depression. WHtR was significantly higher in depressive women (*p* = 0.013) and men (*p* = 0.046) as compared to their non-depressive peers. WC was higher in depressive women (*p* = 0.004) with a similar tendency in men (*p* = 0.15).

In a multiple logistic regression model with WWI, WC, WHtR, age, and sex as independent variables and depression as a dependent variable, only WWI [OR = 1.20 (1.25–1.28)] and female sex [OR = 1.57 (1.31–1.89)] were identified as independent predictors. Consistent results were obtained in both forward and backward regression analyses. These findings indicate that an increase in WWI by 1 cm/√kg raises the probability of depression by 20% while being a woman increases the probability of depression by 57%.

## 4. Discussion

The results of this study indicate that WWI and BMI differ in their potential to discriminate the presence of concomitant diseases in hospitalized older adults. Both high BMI and high WWI appear to identify patients with cardiometabolic diseases such as hypertension, diabetes, and heart failure. WWI, apart from being associated with cardiometabolic conditions, is also related to depression.

In our advanced-age population, both high BMI and high WWI seem to indicate patients with hypertension or diabetes, which is consistent with previous studies [5,15,33,34,35,36,37,38,39,40]. Notably, however, previous studies linking WWI to diabetes mellitus [5,33,34,35] were conducted in younger American and Asian populations. There is scarce research on this issue among European citizens. In contrast to its predictive value for hypertension and diabetes, BMI appears to more effectively identify older patients with dyslipidemia.

In our study, BMI and WWI show a positive association with the prevalence of heart failure in women, with a similar tendency in men. The relationship between BMI-measured obesity and heart failure is well established [41,42]. In a vast analysis including 22,193 individuals with 3062 cardiovascular events from nine prospective follow-up studies, BMI-measured obesity was found to cause coronary heart disease, heart failure, and ischemic stroke [43]. One previous study also found WWI to be a valuable indicator for detecting heart failure occurrence in populations up to 85 years of age (the inclusion criterion) [44]. The present study suggests that this association is valid even in populations up to 105 years old. Furthermore, our study showed a positive relation between WWI and chronic kidney disease in women, which was similarly observed in a prior study in both men and women [45], although it involved a much younger and non-institutionalized population.

The relationship between obesity and osteoporosis is not unequivocal. Some research indicates that patients living with obesity face an increased risk of developing osteoporosis [46], while others suggest that a higher body mass may be a protective factor for bones and diminish the risk of fractures [47]. However, evidence suggests that high body fat percentage negatively impacts bone structure, despite the positive effects of weight on bone strength [48,49]. Additionally, while a higher BMI may protect against certain types of fractures, such as hip, spine, and wrist fractures, it may increase the risk of others, such as ankle fractures [50]. According to our data, osteoporosis is associated with a lower BMI but a higher WWI, particularly in men. Similar results were obtained by Mi et al., who related higher WWI values with degraded bone microarchitecture and concluded that WWI, either alone or in combination with BMI, could serve as a potential screening tool for osteoporosis [51]. This suggests that higher body mass, in general, may provide protection against osteoporosis, while higher body fat may be detrimental.

A lower BMI was found to differentiate patients with diseases typical of advanced age, such as osteoporosis, cancer, or dementia. This finding probably reflects the better functional, cognitive, and nutritional status observed in older adults with a higher BMI, as confirmed by our study. WWI appears to be inversely related to geriatric assessments, with poorer physical function results associated with higher WWI values. In a meta-analysis including 16 cohort studies [52], 13 studies showed an inverse association between BMI and dementia, with 10 of these being statistically significant. Two studies found a positive association between BMI and dementia, while one reported a non-significantly increased risk. According to our data, a higher BMI seems to be a protective factor against dementia, supporting a majority of the existing literature. WWI, however, showed no relationship with dementia, and only a slight negative correlation with the MMSE score in women. A study of the NHANES 2011–2014 population aged 60 years and over found that the risk of cognitive decline, assessed by various tests, may be positively associated with WWI [53,54].

Unlike BMI, WWI was higher in patients with depression, and this association merits further investigation. Depression is one of the most common chronic diseases in the older population. According to the WHO, depression is often underrecognized and undertreated in individuals aged over 65 years [55], which emphasizes the great need to develop better tools that could improve the detection, prevention, and treatment of depression in this age group. Previous studies on obesity and depression have produced inconsistent results, with some of them showing a positive association and others indicating a U-shaped association [56,57]. All of these studies, however, used BMI as the anthropometric measure of obesity. The relation between sarcopenia and depression in older adults is also clear in the literature [58], with some papers reporting the combined effect of obesity and sarcopenia on depression risk [59]. In our study, there was a slight negative correlation between BMI and GDS score in older men, which suggests that a higher BMI may be associated with fewer depressive symptoms in this group. This finding contrasts with data from younger adults, where depression prevalence increases with a higher BMI [5,6]. On the other hand, overweight older men had lower odds of being depressed five years later compared to their normal-weight peers [60].

There is limited research that presents WWI as an indicator for identifying populations at risk for depression. However, a study from the National Health and Examination Nutrition Survey (NHANES) of 2005–2018 delivered research that involved 34,528 adult participants and found a positive relationship between WWI and depression risk [61]. Studies by Liu et al. and Shen et al., based on NHANES 2005–2018 data, also concluded that WWI was positively associated with depressive symptoms [62,63]. Another study conducted by Fei et al. from NHANES showed that WWI was related to higher depression risk, with this relationship being stronger than that of BMI [64]. These studies included non-hospitalized younger and middle-aged adults, whereas our study extends these observations to a much older and vulnerable population. Due to its simplicity, WWI has a great potential for wide clinical application. The mechanism underlying the positive correlation between WWI and depression remains unclear. Our hypothesis is that obesity leads to chronic inflammation, which is associated with a higher risk of depression [65]. Individuals with a higher WWI are more likely to have unhealthy body composition and central obesity [17,18], both of which contribute to chronic inflammation, potentially explaining the positive correlation between WWI and depression symptoms. Additionally, a higher WWI and consequently lower muscle mass [17,18] may result from low physical activity, which is also associated with depression [66,67]. A study by Zhou et al. based on NHANES data also identified an association between a higher WWI and an increased risk of sarcopenia [20]. In accordance with our findings, we observed a positive and significant correlation between WWI and VES-13 and a negative correlation between WWI and ADL and IADL in women, which suggests that individuals with a higher WWI experience greater functional disabilities, limitations in daily activities, and worse self-rated health.

This study has several limitations. Firstly, our sample included neither younger populations nor non-hospitalized patients. Secondly, all the study subjects were Polish; therefore, research covering other ethnic groups is required. On the other hand, a key strength of this study is the large number of participants included in the analysis. Moreover, to the best of our knowledge, this is the first study describing an association between WWI and depression exclusively in hospitalized older adults.

## 5. Conclusions

This study demonstrates an association between a higher WWI and depression, as well as the presence of depressive symptoms assessed by the GDS in hospitalized older adults, both women and men. No such relationship was observed between depression and BMI. Both high BMI and high WWI seem to identify older patients with cardiometabolic diseases such as hypertension and diabetes. Based on these findings, WWI appears to be a promising indicator of depression risk and, similarly to BMI, a useful parameter for the assessment of cardiometabolic risk in older hospitalized adults. However, further studies, particularly those with a prospective design, are required to confirm these results.

## Figures and Tables

**Table 1 nutrients-17-00068-t001:** General characteristic of the study population (*n* = 2945) according to sex.

Variable	All (*n* = 2945)	Women (*n* = 2095)	Men (*n* = 850)	*p*-Value
Age [years]	82 (76–87)	82 (76–87)	82 (76–87)	ns *
Weight [kg]	66 (57–76)	62.5 (54–72)	73 (65–83)	<0.001 *
BMI [kg/m^2^]	25.4 (22.5–28.9)	25.4 (22.3–29.2)	25.5 (22.9–28.1)	ns *
Waist circumference [cm]	90 (81–100)	90 (80–100)	95 (86–103)	<0.001 *
Calf circumference [cm]	34 (31–37)	34 (31–37)	35 (31–38)	<0.001 *
Arm circumference [cm]	27 (24–30)	27 (24–30)	28 (25–30)	0.001 *
WHtR	0.56 (0.51–0.63)	0.57 (0.51–0.63)	0.55 (0.51–0.61)	<0.001 *
WWI [cm/√kg]	11.3 (10.2–12)	11.3 (10.4–12.1)	11.1 (10.3–11.8)	<0.001 *
ADL	6 (4–6)	6 (4–6)	6 (4–6)	ns *
IADL	6 (3–8)	6 (3–8)	7 (3–8)	ns *
GDS	5 (3–8)	5 (3–8)	4 (2–7)	<0.001 *
MMSE	24 (19–28)	24 (19–28)	25 (19.5–28)	ns *
CDT	4 (0–7)	4 (0–7)	5 (0–7)	<0.001 *
VES-13	7 (4–9)	7 (4–9)	6 (3–8)	<0.001 *
NRS	1 (0–2)	1 (0–2)	1 (0–2)	ns *
Hypertension [*n* (%)]	2220 (80.3)	1591 (81)	628 (78)	ns **
Dyslipidemia [*n* (%)]	1208 (43.7)	914 (46.7)	294 (36.5)	<0.001 **
Diabetes [*n* (%)]	743 (26.9)	500 (25.5)	242 (30)	0.04 **
Myocardial infarction [*n* (%)]	304 (11)	168 (8.6)	136 (16.9)	<0.001 **
Ischemic heart disease [*n* (%)]	1028 (37.2)	707 (36.1)	321 (39.8)	ns **
Atrial fibrillation [*n* (%)]	559 (20.2)	365 (18.7)	194 (24.1)	0.01 **
Heart failure [*n* (%)]	1363 (48)	974 (49.7)	389 (48)	ns **
Stroke [*n* (%)]	449 (16.2)	314 (16)	135 (16.7)	ns **
Chronic kidney disease [*n* (%)]	775 (26.3)	550 (26.2)	201 (23.6)	ns **
Pulmonary diseases [*n* (%)]	353 (12.8)	242 (12.3)	111 (13.7)	ns **
Osteoarthritis [*n* (%)]	994 (36)	781 (39.9)	213 (26.4)	<0.001 **
Osteoporosis [*n* (%)]	697 (25.2)	616 (31.4)	80 (9.9)	<0.001 **
Cancer [*n* (%)]	469 (17)	311 (15.8)	158 (19.6)	ns **
Depression [*n* (%)]	917 (33.2)	711 (36.3)	206 (25.5)	<0.001 **
Dementia [*n* (%)]	1052 (28.2)	769 (39.4)	283 (35.3)	ns **
Parkinson’s disease [*n* (%)]	65 (2.2)	34 (1.6)	31 (3.7)	0.01 **

The quantitative values are presented as median and interquartile differences, the qualitative values as numbers and percentages. * Mann–Whitney U-test; ** Chi^2^-test. Abbreviations: BMI—body mass index, WHtR—waist-to-height ratio, WWI—weight-adjusted waist index, ADL—activities of daily living, IADL—instrumental activities of daily living, GDS—geriatric depression scale, MMSE—mini mental state examination, CDT-clock-drawing test, VES-13-vulnerable elders-13 survey, NRS-nutritional risk screening, ns—not significant.

**Table 2 nutrients-17-00068-t002:** Comparison between the weight-adjusted waist index and quantitative variables.

Variable	Women		Men	
	rS	*p*-Value *	rS	*p*-Value *
Age [years]	0.18	<0.001	0.02	ns
Weight [kg]	0.004	ns	0.06	ns
BMI [kg/m^2^]	0.14	<0.001	0.16	<0.001
Waist circumference [cm]	0.7	<0.001	0.74	<0.001
Calf circumference [cm]	0.08	<0.001	0.16	<0.001
Arm circumference [cm]	0.18	<0.001	0.14	<0.001
WHtR	0.79	<0.001	0.79	<0.001
ADL	−0.06	<0.01	0.005	ns
IADL	−0.07	<0.01	−0.03	ns
GDS	0.09	<0.001	0.1	<0.01
MMSE	−0.07	<0.01	−0.007	ns
CDT	−0.005	ns	0.08	<0.05
VES-13	0.13	<0.001	0.04	ns
NRS	−0.004	ns	−0.03	ns

* Spearman rank correlation; Abbreviations: BMI—body mass index, WHtR—waist-to-height ratio, ADL—activities of daily living, IADL—instrumental activities of daily living, GDS—geriatric depression scale, MMSE—mini mental state examination, CDT—clock-drawing test, VES-13—vulnerable elders-13 survey, NRS—nutritional risk score, ns—not significant.

**Table 3 nutrients-17-00068-t003:** Comparison between body mass index and other quantitative variables.

Variable	Women		Men	
	rS	*p*-Value *	rS	*p*-Value *
Age [years]	−0.19	<0.001	−0.22	<0.001
Weight [kg]	0.9	<0.001	0.88	<0.001
WWI [cm/√kg]	0.14	<0.001	0.16	<0.001
Waist circumference [cm]	0.71	<0.001	0.74	<0.001
Calf circumference [cm]	0.55	<0.001	0.16	<0.001
Arm circumference [cm]	0.68	<0.001	0.14	<0.001
WHtR	0.69	<0.001	0.69	<0.001
ADL	0.11	<0.001	0.17	<0.001
IADL	0.16	<0.001	0.17	<0.001
GDS	−0.03	ns	−0.07	<0.05
MMSE	0.19	<0.001	0.15	<0.001
CDT	0.2	<0.001	0.19	<0.001
VES-13	−0.12	<0.001	−0.2	<0.001
NRS	−0.23	<0.001	−0.28	<0.001

* Spearman rank correlation; Abbreviations: WWI—weight-adjusted waist index, WHtR—waist-to-height ratio, ADL—activities of daily living, IADL—instrumental activities of daily living, GDS—geriatric depression scale, MMSE—mini mental state examination, CDT—clock-drawing test, VES—13-vulnerable elders-13 survey, NRS—nutritional risk score, ns—not significant.

**Table 4 nutrients-17-00068-t004:** Weight-adjusted waist index [cm/√kg] for chronic diseases according to sex.

Variable	WWI for Women [Median and Interquartile Difference]			WWI for Men [Median and Interquartile Difference]		
	With Disease	Without Disease	*p*-Value *	With Disease	Without Disease	*p*-Value *
Hypertension	11.39 (10.4–12.2)	10.87 (9.9–11.9)	<0.001	11.18 (10.4–11.8)	10.66 (10–11.5)	<0.001
Dyslipidemia	11.29 (10.4–12.1)	11.31 (10.3–12.1)	ns	11.13 (10.3–11.8)	11.07 (10.3–11.8)	ns
Diabetes	11.67 (10.7–12.4)	11.16 (10.2–12)	<0.001	11.41 (10.5–12)	10.96 (10.2–11.7)	<0.001
Myocardial infarction	11.52 (10.5–12.2)	11.29 (10.3–12.1)	<0.05	11.08 (10.4–11.7)	11.09 (10.2–11.8)	ns
Ischemic heart disease	11.33 (10.3–12.2)	11.28 (10.3–12.1)	ns	11.16 (10.3–11.8)	11.05 (10.3–11.7)	ns
Atrial fibrillation	11.38 (10.4–12.2)	11.29 (10.3–12.1)	ns	11.03 (10.3–11.8)	11.11 (10.2–11.8)	ns
Heart failure	11.43 (10.5–12.2)	11.16 (10.2–12.1)	<0.001	11.12 (10.3–11.8)	11.05 (10.2–11.8)	ns
Stroke	11.39 (10.4–12.1)	11.29 (10.3–12.1)	ns	11.15 (10.3–11.8)	11.07 (10.3–11.8)	ns
Chronic kidney disease	11.68 (10.9–12.5)	11.49 (10.7–12.2)	<0.001	11.39 (10.6–12)	11.35 (10.7–11.9)	ns
Pulmonary diseases	11.38 (10.4–12.1)	11.29 (10.3–12.1)	ns	11.39 (10.6–11.9)	11.05 (10.2–11.8)	<0.05
Osteoarthritis	11.50 (10.5–12.2)	11.14 (10.3–12.1)	<0.001	11.26 (10.7–11.8)	10.96 (10.2–11.8)	<0.01
Osteoporosis	11.32 (10.3–12.2)	11.29 (10.3–12.1)	ns	11.43 (10.8–11.9)	11.04 (10.2–11.8)	<0.01
Cancer	11.19 (10.3–12)	11.32 (10.3–12.1)	ns	11.11 (10.3–11.8)	11.09 (10.3–11.8)	ns
Depression	11.46 (10.5–12.2)	11.19 (10.3–12.1)	<0.001	11.34 (10.5–11.9)	11.01 (10.2–11.7)	<0.001
Dementia	11.36 (10.4–12.2)	11.29 (10.3–12.1)	ns	11.12 (10.2–11.8)	11.09 (10.3–11.8)	ns
Parkinson’s disease	11.61 (10.7–12.2)	11.62 (10.8–12.3)	ns	11.39 (10.5–12)	11.32 (10.5–11.9)	ns

* Mann–Whitney U test; Abbreviations: ns—not significant.

**Table 5 nutrients-17-00068-t005:** Body mass index [kg/m^2^] for chronic diseases according to sex.

Variable	BMI for Women [Median and Interquartile Difference]			BMI for Men [Median and Interquartile Difference]		
	With Disease	Without Disease	*p*-Value *	With Disease	Without Disease	*p*-Value *
Hypertension	25.96 (22.8–29.7)	23.61 (21–26.7)	<0.001	26.06 (23.4–28.6)	24.22 (21.8–26.4)	<0.001
Dyslipidemia	26.37 (23.1–30.1)	24.99 (22–28.3)	<0.001	26.12 (23.8–29.3)	25.26 (22.7–27.7)	<0.001
Diabetes	27.34 (24–32)	24.88 (21.9–28)	<0.001	26.99 (23.8–30.1)	25.26 (22.5–27.7)	<0.001
Myocardial Infarction	25.71 (22.6–29.3)	25.39 (22.3–29.2)	ns	24.8 (22.9–27.4)	25.78 (23–28.4)	ns
Ischemic Heart Disease	25.54 (22.2–29.4)	25.39 (22.5–29.1)	ns	25.51 (22.8–28.1)	25.66 (23–28.1)	ns
Atrial fibrillation	26.32 (23–29.8)	25.32 (22.3–29)	<0.01	25.85 (23.1–28.5)	25.51 (22.9–28)	ns
Heart failure	25.95 (22.6–29.8)	25.1 (22.2–28.3)	<0.001	25.95 (23.2–28.5)	25.28 (22.7–27.9)	ns
Stroke	25.39 (22.3–28.6)	25.4 (22.4–29.3)	ns	25.42 (22.9–27.8)	25.67 (22.9–28.2)	ns
Chronic kidney disease	26.3 (23.2–30)	25.78 (22.2–29.5)	ns	25.67 (22.8–29.2)	26.09 (23.2–29.1)	ns
Pulmonary diseases	25.39 (22.6–29.7)	25.41 (22.3–29.1)	ns	24.93 (22.4–27.4)	25.71 (23–28.1)	<0.05
Osteoarthritis	26.58 (23.4–30.3)	24.84 (21.9–28.2)	<0.001	26.45 (23.7–29.1)	25.35 (22.8–27.8)	<0.01
Osteoporosis	24.97 (22–28.4)	25.71 (22.6–29.4)	<0.001	24.55 (21.9–27.8)	25.71 (23.1–28.1)	<0.05
Cancer	24.68 (22–28.3)	25.56 (22.4–29.3)	<0.05	24.98 (22.2–27.7)	25.73 (23.1–28.4)	<0.05
Depression	25.48 (22.4–29.4)	25.39 (22.3–29)	ns	25.43 (22.6–28.3)	25.65 (23–28.1)	ns
Dementia	24.61 (21.5–27.5)	26.29 (22.9–30.1)	<0.001	24.91 (22.2–27.5)	25.99 (23.2–28.7)	<0.001
Parkinson’s disease	23.91 (21.6–28.7)	26.17 (22.9–30)	ns	26.27 (24.2–29.1)	25.88 (22.9–29.1)	ns

* Mann–Whitney U test; Abbreviations: ns—not significant.

## Data Availability

The data presented in this study are available on request from the corresponding author due to privacy.

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
