# Peer review of "Weight-Adjusted Waist Index (WWI)—A Promising Anthropometric Indicator of Depressive Symptoms in Hospitalized Older Patients"

_nutrients, 2024, doi:10.3390/nu17010068_

Round 1
Reviewer 1 Report
Comments and Suggestions for Authors
Weight-adjusted waist index (WWI) - a promising anthropometric indicator of depressive symptoms in hospitalized older patients.
This is a novel research topic and one that adds important knowledge and understanding of WWI to the literature.
Introduction:
Lines 44-47: Does the following term of normal weight obesity mean something different than sarcopenic obesity? “Large analysis including 177 792 participants suggested that normal 44 weight obesity (defined as normal BMI but high percentage of fat) increases prevalence of 45 cardiometabolic risk factors, but remains undetected with BMI-based screening.
Can it be clarified how normal weight obesity is assessed?
How were participants consented to participate? It indicates those who were excluded, but can you clarify the consent process for this population and any other restrictions on providing consent based on cognitive ability?
Line 141: can more appropriate/scientific terminology be used instead of “Men were heavier and had bigger WC, CC and AC than women”
Discussion, line 219: It would be helpful to specify which cardiometabolic diseases (i.e., listed previously in results), hypertension, diabetes, and osteoarthritis, as this would more accurately reflect which ones.
Line 225: please correct the typo: “There is scares research” should be “scarce.”
Line 239: please adopt person first language, such as “patients living with obesity” vs “obese patients”
Conclusion: it seems this study has findings pertaining to both cardiometabolic disease and depression and it seems the first sentence of the conclusion only highlights depression. Currently reads “This study shows that there is an association between higher WWI and the depression as well as presence of depressive symptoms according to GDS in hospitalized older 310 adults, both in women and men…” It would be more accurate and reflective of this research to include both. Additionally, the rest of the conclusion only mentioned depression, not the other findings. Seems to overly highlight this?
I also wonder if the title of this paper may be misleading as it only mentions depression and not the other cardiometabolic factors? Can you consider expanding the title to highlight the findings directly.
Author Response
This is a novel research topic and one that adds important knowledge and understanding of WWI to the literature.
I would like to sincerely thank you for your valuable feedback and thoughtful comments on my work. Your insights have been extremely helpful in improving the quality and clarity of the manuscript.
Introduction:
Lines 48-47: Does the following term of normal weight obesity mean something different than sarcopenic obesity? “Large analysis including 177 792 participants suggested that normal 44 weight obesity (defined as normal BMI but high percentage of fat) increases prevalence of 45 cardiometabolic risk factors, but remains undetected with BMI-based screening.
Can it be clarified how normal weight obesity is assessed?
Yes, normal weight obesity (NWO) and sarcopenic obesity refer to different conditions, though both are associated with increased health risks, particularly in older adults. Normal weight obesity (NWO) refers to individuals who have a normal BMI (<25 kg/m²) but a high percentage of body fat. In other words, while their weight may not be classified as overweight or obese according to BMI, they still have excess body fat, which can increase the risk of cardiometabolic conditions such as diabetes, hypertension, and heart disease. NWO is often undetected by traditional BMI-based screenings, as BMI does not account for body fat composition. Sarcopenic obesity, on the other hand, refers to a condition where there is both increased body fat and a loss of muscle mass (sarcopenia). This is common in older adults, where the loss of muscle tissue coupled with an accumulation of fat can result in physical frailty and functional decline. Sarcopenic obesity is often associated with poor physical function, increased disability, and higher mortality risk.
So, while both conditions involve abnormal body composition, NWO involves high fat percentage despite a normal BMI, and sarcopenic obesity involves both high fat and low muscle mass, which can lead to greater physical impairments.
It has been specified more precisely in the manuscript Line 49-58:
“Normal weight obesity (NWO), defined as a normal BMI (<25 kg/m2) but with a high percentage of body fat, is an emerging concern in older adults as it leads to physical impairments [8]. The literature findings suggest that older adults with NWO are at an increased risk of sarcopenia, emphasizing the need for more detailed anthropometric assessment as part of screening in this population [9]. A large study involving 177, 792 participants suggests that, though remaining undetected in BMI-based screening, normal weight obesity increases the prevalence of cardiometabolic risk factors [10]. Similarly, sarcopenic obesity, defined as the loss of muscle tissue and muscle strength coupled with an accumulation of body fat, is increasingly prevalent in older populations and exacerbates physical functional impairments and disabilities [11]”
How were participants consented to participate? It indicates those who were excluded, but can you clarify the consent process for this population and any other restrictions on providing consent based on cognitive ability?
Participants who were able to provide informed consent themselves (i.e., those with cognitive ability to understand the study purpose, procedures, and potential risks) were asked to provide written informed consent. Patients who were unable to provide informed consent due to cognitive impairment (such as severe dementia or with other diseases that hindered their verbal and logical contact) were excluded from the study. This information have been added into the manuscript in lines 93-98
Line 141: can more appropriate/scientific terminology be used instead of “Men were heavier and had bigger WC, CC and AC than women”
Has been corrected.
Discussion, line 219: It would be helpful to specify which cardiometabolic diseases (i.e., listed previously in results), hypertension, diabetes, and osteoarthritis, as this would more accurately reflect which ones.
Has been added.
Line 225: please correct the typo: “There is scares research” should be “scarce.”
Has been corrected.
Line 239: please adopt person first language, such as “patients living with obesity” vs “obese patients”
Has been corrected
Line270-273: “The relationship between obesity and osteoporosis is not unequivocal. Some research indicates that patients living with obesity face an increased risk of developing osteoporosis…”
Conclusion: it seems this study has findings pertaining to both cardiometabolic disease and depression and it seems the first sentence of the conclusion only highlights depression. Currently reads “This study shows that there is an association between higher WWI and the depression as well as presence of depressive symptoms according to GDS in hospitalized older 310 adults, both in women and men…” It would be more accurate and reflective of this research to include both. Additionally, the rest of the conclusion only mentioned depression, not the other findings. Seems to overly highlight this?
We have added information regarding the association between WWI and cardiometabolic diseases in the conclusions and abstract, as per the reviewer’s recommendation.
Line 340-348: „This study demonstrates an association between higher WWI and depression, as well as the presence of depressive symptoms as assessed by the GDS, in hospitalized older adults, both in women and men. No such relationship was observed between depression and BMI. Both high BMI and high WWI seem to identify older patients with cardiometabolic diseases such as hypertension and diabetes. Based on these findings, WWI appears to be a promising indicator of depression risk and, similarly to BMI, useful in assessing cardiometabolic risk in older hospitalized adults. Further studies, particularly those with a prospective design, are needed to confirm these results. “
I also wonder if the title of this paper may be misleading as it only mentions depression and not the other cardiometabolic factors? Can you consider expanding the title to highlight the findings directly.
Thank you for your suggestion. After careful consideration, we have decided not to include cardiometabolic diseases in the title. Our intention is to emphasize the novelty of the study in relation to the findings on depression, as the association with cardiometabolic factors is not a new discovery. Therefore, we believe that the current title accurately reflects the main finding of our research.
Reviewer 2 Report
Comments and Suggestions for Authors
In the present study, Korycka-Bloch and col, have evaluated which anthropometric index, BMI or 11 WWI, more accurately associates with prevalence of the most common chronic diseases and com-12 ponents of geriatric assessment in hospitalized older adults.
The Introduction needs an extensive revision, since many phrases are not correctly constructe and the meaning is lacking. In addition, there is no connection among almost of the phrases in introduction. Furthermore, many phrases (affirmations) in the introduction lacks a reference.
The Material and Methods is well written without mistakes.
The Results needs attention for grammar, and specially to construction of the phrases keeping the scientific writting. An example of non-scientific english used is: "Table 1 summarizes general characteristics of the study population according to sex. 140 Men were heavier and had bigger WC, CC and AC than women."...
The Discussion. Although the references used in the Discussion are appropriate, same problem observed in the Introduction remains. So, the Discussion needs an extensive revision, since many phrases are not correctly constructe and the meaning is lacking. In addition, there is no connection among almost of the phrases in Discussion.
So, the manuscript is of a great interest, with high number of participants, but the whole manuscript needs an extensive revision. So my suggestion is Major Revisions.
Comments on the Quality of English LanguageThe entire manuscript needs an extensive english revision.
Author Response
In the present study, Korycka-Bloch and col, have evaluated which anthropometric index, BMI or 11 WWI, more accurately associates with prevalence of the most common chronic diseases and com-12 ponents of geriatric assessment in hospitalized older adults.
I would like to sincerely thank you for your valuable feedback and thoughtful comments on my work. Your insights have been extremely helpful in improving the quality and clarity of the manuscript.
The Introduction needs an extensive revision, since many phrases are not correctly constructe and the meaning is lacking. In addition, there is no connection among almost of the phrases in introduction. Furthermore, many phrases (affirmations) in the introduction lacks a reference.
The introduction has undergone a thorough revision, during which we addressed the issues of sentence construction and clarity. Additionally, we have added appropriate citations and corrected both linguistic and conceptual errors to improve the overall quality and coherence of the text. We believe these changes have significantly enhanced the readability and accuracy of the introduction.
The citations we have added include:
Makovski, T.T.; Schmitz, S.; Zeegers, M.P.; Stranges, S.; van den Akker, M. Multimorbidity and quality of life: systematic literature review and meta-analysis. Ageing research reviews 2019, 53, 100903.
Lynch, D.H.; Petersen, C.L.; Fanous, M.M.; Spangler, H.B.; Kahkoska, A.R.; Jimenez, D.; Batsis, J.A. The relationship between multimorbidity, obesity and functional impairment in older adults. J Am Geriatr Soc 2022, 70, 1442-1449, doi:10.1111/jgs.17683.
Haththotuwa, R.N.; Wijeyaratne, C.N.; Senarath, U. Worldwide epidemic of obesity. In Obesity and obstetrics; Elsevier: 2020; pp. 3-8.
Jia, S.; Liu, L.; Huo, X.; Sun, L.; Chen, X. Association of novel anthropometric indices with all-cause mortality in hypertensive patients: Evidence from NHANES 2007–2018. The Journal of nutrition, health and aging 2024, 28, 100356.
De Lorenzo, A.; Pellegrini, M.; Gualtieri, P.; Itani, L.; El Ghoch, M.; Di Renzo, L. The risk of sarcopenia among adults with normal-weight obesity in a nutritional management setting. Nutrients 2022, 14, 5295.
Petermann‐Rocha, F.; Balntzi, V.; Gray, S.R.; Lara, J.; Ho, F.K.; Pell, J.P.; Celis‐Morales, C. Global prevalence of sarcopenia and severe sarcopenia: a systematic review and meta‐analysis. Journal of cachexia, sarcopenia and muscle 2022, 13, 86-99.
Ozkok, S.; Aydin, C.O.; Sacar, D.E.; Catikkas, N.M.; Erdogan, T.; Bozkurt, M.E.; Kilic, C.; Karan, M.A.; Bahat, G. Sarcopenic obesity versus sarcopenia alone with the use of probable sarcopenia definition for sarcopenia: Associations with frailty and physical performance. Clinical Nutrition 2022, 41, 2509-2516.
Liu, S.; Pan, X.; Chen, B.; Zeng, D.; Xu, S.; Li, R.; Tang, X.; Qin, Y. Association between healthy lifestyle and frailty in adults and mediating role of weight-adjusted waist index: results from NHANES. BMC geriatrics 2024, 24, 757.
Luo, J.; Deng, H.; Wu, Y.; Zhang, T.; Cai, Y.; Yang, Y. The weight-adjusted waist index and frailty: A cohort study from the China Health and Retirement Longitudinal Study. The Journal of nutrition, health and aging 2024, 28, 100322.
Jia, S.; Huo, X.; Liu, L.; Sun, L.; Chen, X. Association of weight-adjusted-waist index with phenotypic age acceleration: Insight from NHANES 2005–2010. The Journal of nutrition, health and aging 2024, 28, 100222.
The Material and Methods is well written without mistakes.
The Results needs attention for grammar, and specially to construction of the phrases keeping the scientific writting. An example of non-scientific english used is: "Table 1 summarizes general characteristics of the study population according to sex. 140 Men were heavier and had bigger WC, CC and AC than women."
We have carefully revised the Results section, addressing both grammar and sentence construction to ensure it aligns with scientific writing standards. The example you mentioned, along with other similar issues, has been corrected accordingly. We believe the revisions have improved the clarity and precision of the section.
The Discussion. Although the references used in the Discussion are appropriate, same problem observed in the Introduction remains. So, the Discussion needs an extensive revision, since many phrases are not correctly constructe and the meaning is lacking. In addition, there is no connection among almost of the phrases in Discussion.
The Discussion section has been thoroughly revised to address the issues you mentioned. We have improved sentence construction, clarified meaning, and ensured better coherence between the phrases. We believe these revisions have significantly enhanced the flow and clarity of the Discussion.
So, the manuscript is of a great interest, with high number of participants, but the whole manuscript needs an extensive revision. So my suggestion is Major Revisions.
Thank you for your feedback and for recognizing the potential of the manuscript. We have carefully addressed the issues you raised, and the manuscript has undergone a thorough language revision by a native speaker. We believe these changes have significantly improved the quality of the manuscript. We hope that the revised version meets your expectations.
Round 2
Reviewer 2 Report
Comments and Suggestions for Authors
The manuscript was enterely reviewed. The criticism presented for the introduction and discussion was solved and now the manuscript permit a fluid reading. As stated before, the manuscript is of a great interest for readers such as geriatricians, cardiologists, intensive medicine, endocrinologists, etc. I believe the methodology presented in the present study will be useful for dozen of scientists and clinicians around the world, since this is an inexpensive method, with a great application and reflecting the presence of depression for hospilazed older patients. So, I recomend the acceptance of this manuscript in its present form.